# MicroRNA-1289 Functions as a Novel Tumor Suppressor in Oral Squamous Cell Carcinoma

**DOI:** 10.3390/cancers15164138

**Published:** 2023-08-17

**Authors:** Koh-ichi Nakashiro, Norihiko Tokuzen, Masato Saika, Hiroyuki Shirai, Nobuyuki Kuribayashi, Hiroyuki Goda, Daisuke Uchida

**Affiliations:** Department of Oral and Maxillofacial Surgery, Ehime University Graduate School of Medicine, Toon 791-0295, Japan; tokuzen@m.ehime-u.ac.jp (N.T.); saika.masato.ll@ehime-u.ac.jp (M.S.); shirai.hiroyuki.li@ehime-u.ac.jp (H.S.); kuri-25@m.ehime-u.ac.jp (N.K.); hiro9832@m.ehime-u.ac.jp (H.G.); udai@m.ehime-u.ac.jp (D.U.)

**Keywords:** oral squamous cell carcinoma (OSCC), microRNA-1289 (miR-1289), magnesium transporter 1 (MAGT1), tumor-suppressive microRNAs

## Abstract

**Simple Summary:**

Tumor-suppressive microRNAs (TS-miRs) play an important role in human malignancies, such as oral squamous cell carcinoma (OSCC). Five such synthetic miRs mimicking human mature miRs were identified in this study. These molecules targeted OSCC cells and significantly reduced their growth. Of these, miR-1289 had the strongest effect, and the administration of an miR-1289 mimic/atelocollagen complex significantly reduced the size of OSCC tumors in vivo. The expression of miR-1289 was also significantly reduced in OSCC tissues. Therefore, identifying genes targeted by miRs, such as magnesium transporter 1 (MAGT1) that is targeted by miR-1289, can be an important therapeutic tool against malignancies like OSCC.

**Abstract:**

Recently, numerous tumor-suppressive microRNAs (TS-miRs) have been identified in human malignancies. Here, we attempted to identify novel TS-miRs in oral squamous cell carcinoma (OSCC). First, we transfected human OSCC cells individually with 968 synthetic miRs mimicking human mature miRs individually, and the growth of these cells was evaluated using the WST-8 assay. Five miR mimics significantly reduced the cell growth rate by less than 30%, and the miR-1289 mimic had the most potent growth inhibitory effect among these miRs. Subsequently, we assessed the in vivo growth-inhibitory effects of miR-1289 using a mouse model. The administration of the miR-1289 mimic–atelocollagen complex significantly reduced the size of subcutaneously xenografted human OSCC tumors. Next, we investigated the expression of miR-1289 in OSCC tissues using reverse transcription–quantitative PCR. The expression level of miR-1289 was significantly lower in OSCC tissues than in the adjacent normal oral mucosa. Furthermore, 15 genes were identified as target genes of miR-1289 via microarray and Ingenuity Pathway Analysis (IPA) microRNA target filtering. Among these genes, the knockdown of magnesium transporter 1 (MAGT1) resulted in the most remarkable cell growth inhibition in human OSCC cells. These results suggested that miR-1289 functions as a novel TS-miR in OSCC and may be a useful therapeutic tool for patients with OSCC.

## 1. Introduction

Oral squamous cell carcinoma (OSCC) is the most frequently occurring cancer among head and neck squamous cell carcinomas, with an estimated incidence of 377,713 new cases and 177,757 deaths in 2020 worldwide [1]. Despite increasing knowledge of OSCC pathogenesis and advances in chemotherapy, radiotherapy, and surgery, the five-year survival rate of patients with OSCC remains at approximately 70% [2]. Therefore, a better understanding of the pathogenesis of OSCC is required to develop optimal therapeutic approaches.

Human malignancies including OSCC are caused by genomic and epigenomic alterations. In 2011, whole exome analysis first showed the landscape of gene mutations in squamous cell carcinoma of the head and neck, including the oral cavity [3,4]. Mutations in tumor-suppressor genes, such as TP53, CDKN2A, and NOTCH1, were found at high frequencies, and activating mutations in oncogenes PIK3CA and HRAS, which are effective targets for molecular-targeted drugs, were each detected at less than 10%. Therefore, we previously attempted to identify useful target molecules for the treatment of OSCC using microarray analysis, and we identified several cancer-related genes, such as AKT1, AURKA, RRM2, and CDCA5, that were commonly overexpressed in human OSCC cell lines and tissues [5,6,7,8].

MicroRNAs (miRs) have attracted more attention than other classes of non-coding RNAs over the past several years, especially because of their essential role in cancer development. More than 50% of the known miRs have been shown to participate in tumorigenesis and/or metastasis by directly targeting oncogenes or tumor-suppressor genes [9,10]. A number of miRs have been shown to be dysregulated in OSCC and to function as oncogenic miRs (OncomiRs) [11]. For example, miR-21, known as an OncomiR, is overexpressed in OSCC tissues and supports the malignant phenotypes of human OSCC cells. It reduces the expression of several tumor-suppressor genes, including programmed cell death 4 (PDCD4) and reversion-inducing cysteine-rich protein with Kazal motifs (RECK) [12,13]. We also previously reported miR-361-3p as an OncomiR candidate in OSCC [14]. In contrast, tumor-suppressive miRs (TS-miRs), such as miR-34a, miR-145, and miR-206, have also been found in OSCC [15,16,17]. Furthermore, the therapeutic efficacy of miR-34a mimics in patients with advanced solid tumors has been reported [18]. Therefore, in the present study, we attempted to identify novel TS-miRs in OSCC using function-based screening.

## 2. Materials and Methods

### 2.1. Cells and Cell Culture

We used three human OSCC cell lines, green fluorescent protein (GFP)-SAS [19], Ca9-22, and HSC2 [20]. GFP-SAS cells were established as described previously [19]. Ca9-22 and HSC2 cells were provided by the RIKEN BRC through the National Bio-Resource Project of the MEXT/AMED, Japan. All cell lines were maintained in Dulbecco’s modified Eagle medium (DMEM; Fujifilm Wako, Osaka, Japan) supplemented with 10% fetal bovine serum (FBS; Thermo Fisher Scientific, Waltham, MA, USA), 100 U/mL penicillin, and 100 μg/mL streptomycin (Fujifilm Wako), referred to here as “complete medium”. Cells were grown in an incubator with a humidified atmosphere of 95% air and 5% CO_2_ at 37 °C. The identity of these cells was verified using short tandem repeat (STR) profiling (Bex, Tokyo, Japan).

### 2.2. Samples

Twenty OSCC tissue samples were obtained from patients at the Ehime University Hospital between July 2012 and June 2017. Tissues were collected from resected specimens of primary tumors (12 men and 8 women; average age 65.6 years old). The Institutional Review Board (IRB) of the Ehime University Hospital approved this study (1206017).

### 2.3. Transfection with Human Mature miR Mimics and Small Interfering RNAs (siRNAs)

Functional screening for TS-miRs in human OSCC cells was performed using the human mature miR mimics library version 13.0 (Hokkaido System Science, Sapporo, Japan). Human mature miR-1289 mimics and miR-non-target (miR-NT) for in vitro and in vivo assay were purchased from Sigma-Aldrich (St. Louis, MO, USA). Silencer Select siRNAs for specific genes and Silencer Select Negative Control (siNT) were purchased from Thermo Fisher Scientific. Transfection was performed using Lipofectamine RNAiMAX (Thermo Fisher Scientific) mixed with 20 nM miR mimics or 10 nM siRNAs for the cell growth assay, reverse transcription–quantitative PCR (RT-qPCR), and microarray.

### 2.4. Cell Growth Assay

Cells (2 × 10^3^/well for GFP-SAS, 3 × 10^3^/well for Ca9-22 and HSC2) were seeded into 96-well plates (Corning, Corning, NY, USA) in complete medium with human mature miR mimics or siRNAs and 0.2% Lipofectamine RNAiMAX with a final volume of 100 μL. After 72 h, the cell count was evaluated using the WST-8 assay (Cell Counting Kit-8; Dojindo, Kumamoto, Japan).

### 2.5. Total RNA Extraction

Total RNA was extracted by lysing OSCC cells and tissues after homogenization using a TissueLyser (Qiagen, Hilden, Germany) in ISOGEN (Nippon Gene, Tokyo, Japan) according to the manufacturer’s protocol. RNA concentration was determined using a UV spectrophotometer (GeneQuant; GE Healthcare Bioscience, Piscataway, NJ, USA). RNA quality was confirmed using an Agilent 2100 Bioanalyzer (Agilent Technologies, Santa Clara, CA, USA), and the samples were stored at −80 °C until use.

### 2.6. Microarray and Prediction of miR-1289 Target Genes

We utilized 500 ng of total RNA to generate biotin-labeled cRNA using the Affymetrix GeneChip^®®^ 3′ IVT PLUS Reagent Kit (Thermo Fisher Scientific). Biotin-labeled RNA was hybridized to Affymetrix Human Genome U-219 Array Strips (Thermo Fisher Scientific), according to the manufacturer’s instructions. After washing and staining the array strips, the signal was developed and scanned using an Affymetrix GeneAtlas system (Thermo Fisher Scientific). We analyzed the results using GeneSpring GX 14 (Agilent Technologies) and an Ingenuity Pathway Analysis (IPA) microRNA Target Filter (Qiagen). Microarray data were deposited in the Gene Expression Omnibus (GEO, experiment number: GSE227010) according to the minimum information about microarray experiment (MIAME) guidelines.

### 2.7. RT-qPCR

The relative quantity of magnesium transporter 1 (MAGT1) mRNA was determined using SYBR Green and the comparative threshold cycle (Ct) method (ΔΔCt method). Hydroxymethylbilane synthase (HMBS) was used as an internal control. PCR amplification was performed to assess mRNA levels using a one-step RT-qPCR reagent (One-Step SYBR^®®^ PrimeScript RT-PCR Kit II (Perfect Real Time); Takara, Kusatsu, Japan) based on the manufacturer’s protocol. The thermal cycling conditions were as follows: RT at 42 °C for 5 min and 95 °C for 10 s, followed by 40 cycles at 95 °C for 5 sec and 60 °C for 30 sec. SYBR Green I fluorescence was detected using ViiA^TM^7 (Thermo Fisher Scientific). The primer sequences used were as follows: MAGT1, forward 5′-GGG ATT GCT TTT GGC TGT TA-3′ and reverse 5′-TAT GGG CAT ATG GTG GTC CT-3′; HMBS, forward 5′-CAT GCA GGC TAC CAT CCA TGT C-3′ and reverse 5′-GTT ACG AGC AGT GAT GCC TAC CAA-3′.

To assess mature miR-1289 expression levels, we used a miScript PCR system (miScript II RT Kit, miScript SYBR Green PCR Kit, and miScript Primer Assays; Qiagen) following the manufacturer’s instructions. The relative miR levels were quantified using the comparative Ct method (ΔΔCt method). The assay includes two steps: RT and qPCR. RT was performed using a thermal cycler (GeneAmp^®®^ PCR System 9700; Thermo Fisher Scientific). The reaction conditions were as follows: 37 °C for 60 min and 95 °C for 5 min. PCR assays were performed using ViiA^TM^7. Parameters were 95 °C for 15 min, followed by 40 cycles at 94 °C for 15 s, 55 °C for 30 s, and 70 °C for 30 s. The PCR results were recorded as Ct values and normalized against RNU6B. Specific primer for hsa-miR-1289 (Cat. #MS00014511, Qiagen) and the endogenous control RNU6B (Cat. #MS00033740; Qiagen) were used.

### 2.8. Luciferase Reporter Assay

The plasmid, miTarget miRNA 3′ Target Clone (Cat. #HmiT020443-MT01, GeneCopoeia, Rockville, MD, USA) was specifically synthesized and used for a luciferase reporter assay. These plasmids contain firefly luciferase that is fused to the 3′-untranslated region (UTR) of human MAGT1 and *Renilla* luciferase, which functions as a tracking gene. GFP-SAS cells were seeded in 96-well plates. After 24 h, the cells were cotransfected with miR-1289 mimics or miR-NT (100 nM) and a plasmid containing MAGT1 3′-UTR (10 ng/well) complexed with Lipofectamine 3000 (Thermo Fisher Scientific), and incubated for 24 h. Firefly and *Renilla* luciferase activities were measured sequentially using the Dual-Glo^®®^ luciferase Assay System (Promega, Madison, WI, USA). The results are expressed as relative luciferase activity units using Wallac 1420 ARVO MX/Light (PerkinElmer, Waltham, MA, USA). The activities were normalized to *Renilla* luciferase activity.

### 2.9. Xenograft Model

GFP-SAS cells (2 × 10^6^) complexed with matrigel (BD, Flanklin Lakes, NJ, USA) in 100 µL aliquots were injected subcutaneously at two sites in the flanks of male athymic nude mice (CLEA Japan, Tokyo, Japan). Two weeks later, the tumor-bearing nude mice were randomly divided into the following treatment groups: miR-1289 mimics or miR-NT. The final concentration of miRs was 100 ng in 100 µL miRs/atelocollagen (AteloGene; Koken, Tokyo, Japan) complexes. These complexes were injected around the tumor every 7 days. Tumor diameters were measured at regular intervals using digital calipers, and tumor volume (mm^3^) was calculated using the following formula: length × width × height × 0.523. Fifteen days after the first administration of miRs, the GFP-SAS xenografts were dissected. All animal experiments were approved by the Ehime University Animal Care Committee (05MA-19-14).

### 2.10. Statistical Analysis

All in vitro experiments were performed in triplicate and repeated thrice. Student’s *t*-test was used to determine the significance of the differences between groups. Differences with *p* values of less than 0.05 were considered statistically significant.

## 3. Results

### 3.1. Identification of TS-miR Candidates in Human OSCC Cells

In comprehensive functional analysis using 968 human mature miR mimics (Appendix A), 5 miRs inhibited cell proliferation by more than 70% compared to the control. Among them, miR-1289 was the most potent inhibitor of the growth of human OSCC cells, GFP-SAS (Figure 1A and Appendix A). Subsequently, we transfected a human mature miR-1289 mimic at a concentration of 20 nM into other human OSCC cell lines, Ca9-22 and HSC2. We found that miR-1289 had a significant growth inhibitory effect on all the human OSCC cells tested (Figure 1B).

### 3.2. Target Genes of miR-1289 in Human OSCC Cells

To gain further insight into which genes are regulated by miR-1289, we performed gene expression analysis following miR-1289 mimic transfection and compared it with negative controls in human OSCC cell lines (GFP-SAS, Ca9-22, and HSC2). The expression of 31 genes was downregulated by more than two-fold in miR-1289 mimic transfectants in all human OSCC cells. Among these genes, 15 genes had putative target sites of miR-1289 in their 3′-UTR (Table 1), as shown by the IPA microRNA Target Filter. To clarify the function of these genes in the proliferation of human OSCC cells, we transfected GFP-SAS, Ca9-22, and HSC2 cells with 10 nM siRNAs specific to each gene. Knockdown of MAGT1, GINS1, DDAH1, or BECN1 significantly inhibited cell growth in all human OSCC cell lines (Figure 2A–C). Among them, MAGT1 suppression resulted in remarkable inhibition of cell growth in human OSCC cell lines.

### 3.3. MAGT1 as a Direct Target Gene of miR-1289 in Human OSCC Cells

Transfection with the mature miR-1289 mimic decreased MAGT1 mRNA expression in GFP-SAS cells (Figure 3A). Furthermore, TargetScan indicates that the 3′-UTR of MAGT1 has three binding sites for miR-1289 (Figure 3B). To test the hypothesis that MAGT1 is a target of miR-1289, a reporter plasmid harboring the 3′-UTR of MAGT1 downstream of the luciferase-coding region was used. MAGT1 has three target sites for miR-1289. GFP-SAS cells were co-transfected with this reporter plasmid and mature miR-1289 mimic. Consequently, the miR-1289-transfected group significantly reduced luciferase activity (Figure 3C).

### 3.4. Effect of miR-1289 Mimics on the In Vivo Growth of Human OSCC Cells

We assessed the in vivo growth-inhibitory effects of the miR-1289 mimic in a mouse xenograft model. We selected GFP-SAS cells for the in vivo assay, because only these cells showed stable tumorigenicity among the OSCC cells used. We found that, compared to the control group, the miR-1289 mimic/atelocollagen complex group showed a significant reduction in the size and weight of the subcutaneously xenografted GFP-SAS tumors (Figure 4A–C). Furthermore, the administration of miR-1289 mimic/atelocollagen complexes suppressed the expression of MAGT1 mRNA in xenografted GFP-SAS tumors (Figure 4D). During the administration of miR-1289 mimics, no reductions in food intake and body weight were observed in the mice. Furthermore, no obvious invasion or metastasis of tumor cells to other organs was observed when sacrificed.

### 3.5. Expression of miR-1289 in OSCC Tissues

RT-qPCR showed that the expression levels of mature miR-1289 were significantly downregulated in 20 OSCC tissue samples compared to those in their adjacent normal oral mucosa tissues (Figure 5A). Subsequently, we evaluated the expression levels of MAGT1 in OSCC tissues using our previous microarray analysis (GSE36090). Upregulation of MAGT1 expression was observed in OSCC tissues compared to those in adjacent normal oral mucosa tissues (Figure 5B).

## 4. Discussion

MicroRNAs are a new class of small RNAs that regulate the expression of many genes, and numerous studies have revealed the aberrant expression of miRs in human malignancies [10]. Various TS-miRs have been reported in many types of cancers [21,22,23,24,25]. However, miR expression profiles in OSCC have been reported in only a few studies [26]. An increased understanding of the expression and function of miRs involved in OSCC development and progression would help improve its diagnosis and treatment.

We attempted to identify novel TS-miRs in human OSCC cells using a mature human miR mimic library and observed that five miR mimics remarkably suppressed the growth of these cells by more than 70%. We showed that miR-1289 mimics potently suppressed the growth of human OSCC cells. However, to the best of our knowledge, there are only two published reports on the biological function of miR-1289 and none in the field of oncology, including OSCC. The first report showed that miR-1289 directly binds to a zipcode-like 25 nucleotide sequence in the 3′-UTR of mRNAs and orchestrates the transfer of these mRNAs into microvesicles [27]. The second report indicated that miR-1289 was upregulated by *Helicobacter pylori* infection and directly targeted the H-K-ATPase α-subunit in human gastric epithelial cells [28].

In this study, we showed that the restoration of miR-1289 expression in human OSCC cells significantly inhibited their growth in vitro and in vivo. Furthermore, downregulation of miR-1289 expression was observed in OSCC tissues. Subsequently, using microarray analysis and an IPA microRNA Target Filter, we identified 15 target gene candidates for miR-1289 in human OSCC cells. Among them, targeting MAGT1, GINS1, DDAH1, or BECN1 by siRNAs significantly suppressed the growth of all human OSCC cell lines. MAGT1 is a mammalian Mg^2+^-selective transporter [29] and has oncogenic functions in breast cancer, cervical cancer, and glioma [30,31,32]. GINS1 is essential for maintaining genome stability and ensuring accurate DNA replication. In glioma cells and tissues, GINS1 promoted cell proliferation and migration through ubiquitin-specific protease 15-mediated deubiquitination of TOP2A protein, and its high expression predicted an advanced clinical grade and a poor survival [33]. DDAH1 is a key enzyme that metabolizes asymmetric dimethylarginine, thereby regulating nitric oxide production. Small-molecule DDAH1 inhibitors significantly suppressed the formation of capillary-like tube structures by human breast cancer cells [34]. BECN1 plays a critical role in regulating the process of autophagy. In colorectal cancer, IL-6 induced autophagy through the activation of BECN1 and promoted chemotherapy resistance [35]. Replacement of miR-1289 can target these multiple genes, suggesting that it may represent a novel therapeutic approach for human malignancies including OSCC.

Therapeutic strategies for miR replacement have recently been evaluated in interventional clinical trials. The first miR-based therapy was an miR-16 mimic for patients with malignant pleural mesothelioma [36]. Another therapy is the use of miR-34a mimic for patients with advanced solid tumors [18]. These reports demonstrate the therapeutic efficacy of miR mimics in human malignancies. However, serious adverse events owing to off-target immune activation have also been reported [37]. Therefore, the RNA modifications used in mRNA vaccines and/or changes in the administration method are necessary for miR replacement therapy. OSCC originates in the oral cavity and metastasizes to the neck lymph nodes, making it possible to directly inject miR-1289 mimics into tumors. The local administration of miR-1289 mimics may be an attractive treatment option for patients with advanced or inoperable OSCC.

## 5. Conclusions

Downregulation of miR-1289 expression is frequent in human OSCC tissues. The replacement of miR-1289 markedly suppressed the growth of human OSCC cells with MAGT1 suppression in vitro and in vivo, suggesting that miR-1289 functions as a novel TS-miR in OSCC. The miR-1289-regulated novel cancer pathways could provide new insights into the molecular mechanisms of OSCC and contribute to the development of new therapeutic strategies for patients with OSCC.

## Figures and Tables

**Figure 1 cancers-15-04138-f001:**
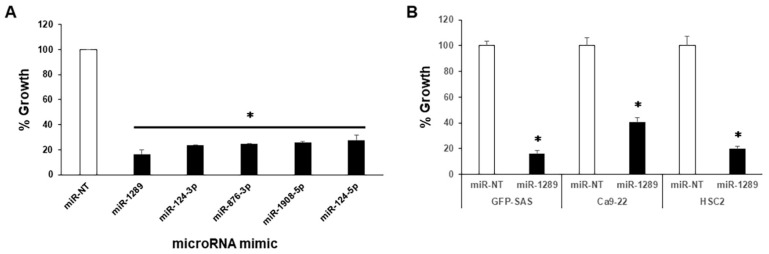
Identification of TS-miR candidates in human OSCC cells. (**A**) Percentage growth of human OSCC cells, GFP-SAS, 72 h after transfection with 968 human mature miR mimics at a concentration of 20 nM each (evaluated by WST-8 assay). (**B**) Percentage growth of other human OSCC cells, Ca9-22 and HSC2, after transfection. All cells were significantly suppressed by miR-1289. Bars denote the standard deviations (SDs) of samples analyzed in triplicate. *, *p* < 0.01 compared to control culture. miR-NT, miR-non target.

**Figure 2 cancers-15-04138-f002:**
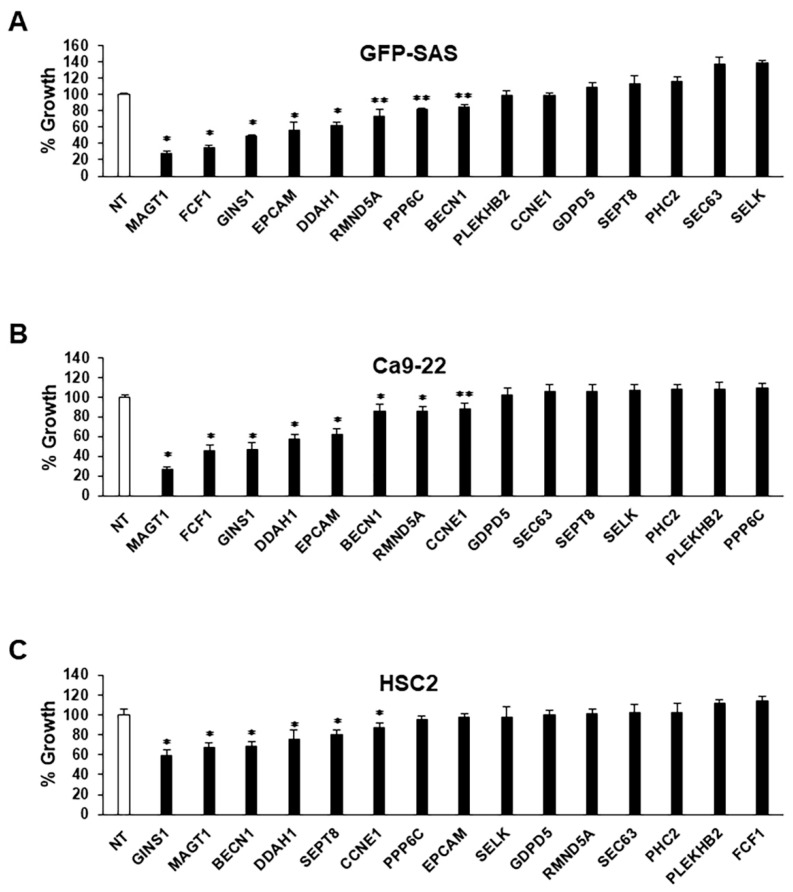
Knockdown of target genes of miR-1289 in human OSCC cells. Percentage cell growth of (**A**) GFP-SAS, (**B**) Ca9-22, and (**C**) HSC2 cells 72 h after transfection with 10 nM siRNAs specific for each target gene. Cell growth was evaluated using the WST-8 assay. Bars denote the SDs of samples analyzed in triplicate. *, *p* < 0.01, **, *p* < 0.05 compared to control culture. NT, non target.

**Figure 3 cancers-15-04138-f003:**
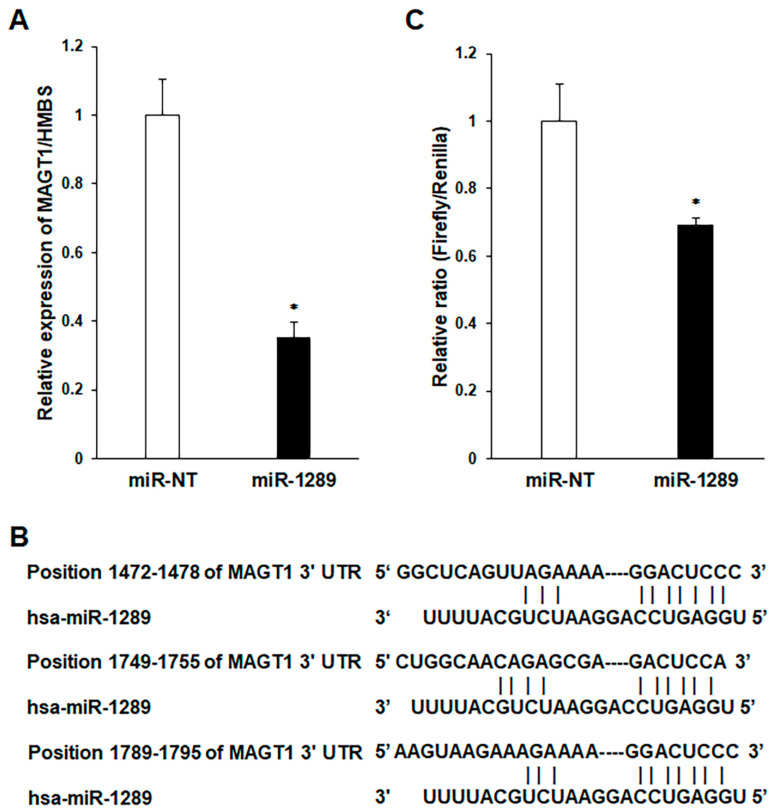
miR-1289 directly targets MAGT1. (**A**) MAGT1 mRNA expression levels after transfection of miR-1289 mimics in GFP-SAS cells. (**B**) Three binding sites of miR-1289 in the 3′-UTR of MAGT1. (**C**) Results of luciferase reporter assay for transfected cells. The miR-1289 mimics downregulated MAGT1 mRNA expression and significantly reduced luciferase activity. Bars denote the SDs of samples analyzed in triplicate. *, *p* < 0.01 compared to control culture. miR-NT, miR-non target.

**Figure 4 cancers-15-04138-f004:**
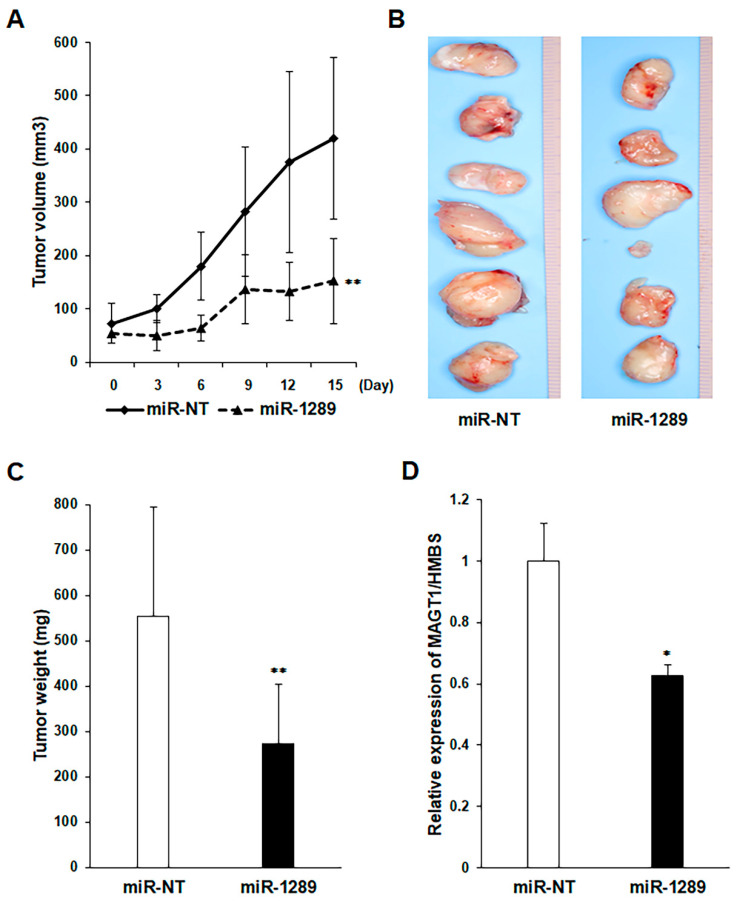
miR-1289 mimics affect tumor cell growth in vivo. (**A**) Tumor volume measured every 3 d during administration of miR-1289 mimics to GFP-SAS tumor cells. (**B**) Resected tumor images and (**C**) tumor weight 15 days after the first administration of miRs. Both tumor volume and weight were reduced in the miR-1289 mimics group compared with that in the control group. (**D**) miR-1289 mimics also suppressed the expression of MAGT1 mRNA. Bars denote the SDs of six tumors in each analyzed group. *, *p* < 0.01, **, *p* < 0.05 compared to control group. miR-NT, miR-non target.

**Figure 5 cancers-15-04138-f005:**
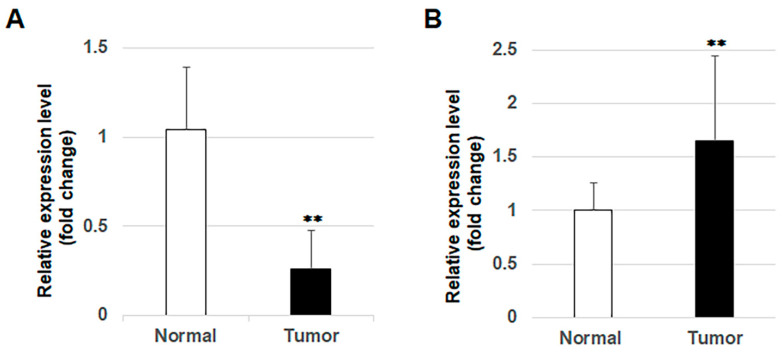
Expression of miR-1289 and MAGT1 in OSCC tissues. (**A**) Expression levels of miR-1289 were significantly downregulated in the 20 OSCC tissues compared to in 20 adjacent normal oral mucosa tissues, evaluated using RT-qPCR. (**B**) Expression levels of MAGT1 were significantly upregulated in the 10 OSCC tissues compared to in 3 adjacent normal oral mucosa tissues in our previous microarray analysis (GSE36090). Bars denote the SDs of the samples in each analyzed group. **, *p* < 0.05 compared to control.

**Table 1 cancers-15-04138-t001:** Downregulated genes from transfection of miR-1289 mimics in human OSCC cells.

Gene Symbol	Gene Name	GFP-SAS (FC) ^1^	Ca9-22 (FC) ^1^	HSC2 (FC) ^1^
SEPT8	septin 8	−3.56	−6.18	−3.90
CCNE1	cyclin E1	−3.54	−8.93	−2.27
FCF1	FCF1 small subunit processome component homolog	−3.20	−3.99	−3.03
SELK	selenoprotein K	−3.16	−4.42	−3.30
DDAH1	dimethylarginine dimethylaminohydrolase 1	−2.48	−2.85	−5.87
RMND5A	required for meiotic nuclear division 5 homolog A	−2.39	−2.41	−2.30
MAGT1	magnesium transporter 1	−2.35	−4.78	−2.89
GINS1	GINS complex subunit 1	−2.21	−4.18	−2.88
PHC2	polyhomeotic homolog 2	−2.21	−3.62	−2.66
BECN1	beclin 1, autophagy related	−2.17	−2.20	−2.15
PPP6C	protein phosphatase 6, catalytic subunit	−2.13	−2.81	−2.47
EPCAM	epithelial cell adhesion molecule	−2.12	−3.88	−3.27
GDPD5	glycerophosphodiester phosphodiesterase domain containing 5	−2.11	−2.34	−3.12
SEC63	SEC63 homolog	−2.08	−2.37	−2.17
PLEKHB2	pleckstrin homology domain containing, family B member 2	−2.07	−2.34	−2.26

^1^ FC, Fold change.

## Data Availability

Data supporting the reported results can be found at https://www.ncbi.nlm.nih.gov/geo/query/acc.cgi?acc=GSE227010, released on 30 June 2023.

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
