# Peer review of "MicroRNA-1289 Functions as a Novel Tumor Suppressor in Oral Squamous Cell Carcinoma"

_cancers, 2023, doi:10.3390/cancers15164138_

Round 1

Reviewer 1 Report

The study aimed to prove that miR-1289 could function as a novel TS-miR and be a useful therapeutic tool for patients with OSCC.

This paper is innovative, but there are still shortcomings that need to be revised:

1. Lack of tumor images in vivo.

2. Lack of research on molecular mechanisms related to MAGT1.

3. Lack of MAGT1 expression detection at the tissue level.

fine

Author Response

We thank the reviewer for the points.

  1. We added the tumor images in vivo in Fig. 4B.
  2. Since the molecular mechanisms of MAGT1 in human malignancies is largely unknown, it will be a subject of next research.
  3. We added the expression levels of MAGT1 in OSCC tissues in Fig. 5B.

Reviewer 2 Report

tumor suppressive miRs play an important role in OSCC,miR1289 is protumorigenic, and anti miR1289 reduced OSCC, among targeted genes a magnesium transporter has an important role.

miR1289 causes a downregulation of several genes including Septin 8,EPCAM and SEC63

The authors suggested that miR-1289 functions as a novel TS-miR in OSCC and may be a useful therapeutic tool for patients 32 with OSCC. 

Author Response

We thank the comments by the reviewer.

Reviewer 3 Report

Tha manuscript from Nakashino et al. describes the function of a tumor suppressor miR-1289 in oral squamous cell carcinoma.

The results of this manuscript are very interesting and clearly presented. They might have some therapeutic applications in the treatment of this cancer type.

However, I have two small observations that I would like to be considered:

1. The introduction is too short. A paragraph on the molecular mechanisms driving this cancer type might be included.

2. Why did the authors not use an RNA-seq approach to find the target genes for miR-1289?

Author Response

We thank the reviewer the points.

  1. We added the paragraph regarding the molecular mechanisms of OSCC progression in the introduction section (line 44-52).
  2. RNA-seq is expensive and we have a microarray system in our laboratory.

Reviewer 4 Report

The authors studied the tumor suppressor role of miRNA-1289 in oral squamous cell carcinoma using OSCC cell lines and tissues. Overall, the manuscript is well written with a clear goal of the study.

- Fig.2: The TS-miRs/siRNA sequence information is missing in the materials/methods. It should be more detailed.

- Fig. 3: 3'-UTR and target sites of miR-1289 could be on Fig.3 with more clear sequence information.

- Fig. 4:

1) if possible, it would be nice to show a representative picture of these tumor volume/size changes in Fig.4.

2) Is there any evidence of no cancer cells migration/metastasis to other organs? It would be better to show other indicators such as total body weight loss or imaging data, if possible.

3) It would be nice to see the MAGT1 protein levels in C.

- Fig. 5: deltaCt is calculated as Ct(a reference gene)-Ct(a target gene) and this is not correct. ΔCt = Ct(a target gene)−Ct(a reference gene). It would be better to be presented as ΔΔCt (delta delta Ct or also known as the 2–∆∆Ct method), which shows a fold change.

- In Fig.5, What are the MAGT1 levels in these 20 OSCC tissues?

Author Response

We thank the reviewer the points.

  1. TS-miR sequences are indicated in Table S1. Synthetic siRNAs used in this study are commercially available, but their sequences have not been opened.
  2. We added the target sites of miR-1289 in MAGT1 3'-UTR in Fig. 3B
  3. 1) We added the tumor images in vivo in Fig. 4B. 2)The description regarding migration/metastasis and body weight loss were added (line 216-218). 3)The antibodies suitable for western blotting were not available.
  4. We showed the relative expression levels (fold change) by ΔΔCt method in Fig. 5A.
  5. We added the expression levels of MAGT1 in OSCC tissues in Fig. 5B.

Reviewer 5 Report

The Authors suggested that 31 miR-1289 functions as a novel TS-miR in OSCC and may be a useful therapeutic tool for patients 32 with OSCC. 

However, some concepts are very unclear.

Usually there is an inverse correlation between miRNAs expression and gene expression. Therefore my questions are:

- how is possible that miR1289 decreased gene expression if itself is decreased in OSCC tissues (Figure 5)? There is a loop between miRNA-gene-miRNA?

- how reduced expression of EPCAM, BECN1, GDPD5 might favour cell growth? These molecules usually favour cell proliferation

Something did not run well, please better explain.

Discussion shlud be expanded.

English should be improved.

Please add in vivo images not only graphs.

English should be improved.

Author Response

We thank the reviewer for the points.

  1. Since the regulation of miR-1289 expression in OSCC is largely unknown, it will be a subject of next research.
  2. Knockdown of EPCAM and BECN1 suppressed the growth of human OSCC cells. MiR-1289 can function as a TS-miR by targeting these genes that support the cell growth. Knockdown of GDPD5 had no effect on the growth of human OSCC cells.
  3. We added the explanation (line 184-186).
  4. We added the description in the discussion section (line 257-271).
  5. English language editing by Editage.
  6. We added the tumor images in vivo in Fig. 4B.

Round 2

Reviewer 4 Report

The authors adequately addressed my concerns.